# Synthetic Pseudo-Spin-Hall effect in acoustic metamaterials

Matthew Weiner [1,2,3,4], Xiang Ni [1,2,5], Andrea Alù [1,2,5] &
Alexander B. Khanikaev [1,2,3] ✉

While vector fields naturally offer additional degrees of freedom for emulating spin, acoustic pressure field is scalar in nature, and it requires engineering of synthetic degrees of freedom by material design. Here we experimentally demonstrate the control of sound waves by using two types of engineered acoustic systems, where synthetic pseudo-spin emerges either as a consequence of the evanescent nature of the field or due to lattice symmetry. First, we show that evanescent sound waves in perforated films possess transverse angular momentum locked to their propagation direction which enables their directional excitation. Second, we demonstrate that lattice symmetries of an acoustic kagome lattice also enable a synthetic transverse pseudo-spin locked to the linear momentum, enabling control of the propagation of modes both in the bulk and along the edges. Our results open a new degree of control of radiation and propagation of acoustic waves thus offering new design approaches for acoustic devices.

In recent years, synthetic degrees of freedom (SDOFs) have significantly expanded the landscape of classical physics by enabling the emulation of relativistic and topological phenomena. Dirac and Weyl physics[1–6] and a broad range of topological phases of matter have been successfully realized in artificial photonic and mechanical materials, facilitating an unprecedented control over propagation and scattering of waves. Artificial electromagnetic and acoustic materials with designer properties, referred to as metamaterials, and endowed with synthetic spin degrees of freedom have made a transformative impact on materials sciences and engineering by enabling new approaches to scatter, trap, and radiate fields. Besides the purely applied interest in new technologies, such metamaterials also have demonstrated unparalleled potential for exploring and testing some of the most fascinating and advanced scientific concepts in tabletop experimental systems. Klein tunneling in Dirac-like systems[7], Weyl points and Fermi arcs[8–14], as well as topological edge, surface[15–18], and higher-order boundary states[19–30] have been widely explored in the past few years using systems with deliberately designed geometric and material parameters to emulate the respective effective Hamiltonians. A key ingredient to unveil this new physics of SDOFs are synthetic gauge potentials acting on them[31–33]. While the

introduction of additional symmetries, such as sublattice, duality, or crystalline symmetries, the manyfold degeneracies emulating components of SDOFs can be engineered, deliberate symmetry reductions can further be used to controllably remove some of these degeneracies thus emulating pseudo-magnetic (or synthetic gauge) fields acting on pseudo-spins. In the context of acoustics, this approach was successfully used to demonstrate Zak phase in SSH acoustic lattices[34], valley-Hall effect[33], and Dirac cones and Weyl points and related topological phenomena in 3D acoustic structures[8,12,14,35,36], and more recently higher order topological phases[22,23,25,26,28–30]. Here we show that pseudo-spins, which can naturally arise for evanescent fields or engineered via symmetries, can be used to control radiation by its guiding and steering in acoustic lattices with coupled angular and linear momenta. Here we experimentally demonstrate pseudo-spin-Hall effect in two types of metamaterial systems exhibiting such coupling.

## Results

### Pseudo-Spin-Hall effect of evanescent acoustic fields

It is known that the electric field of surface plasmon-polaritons possesses elliptical polarization with handedness locked to their propagation

[1]Department of Electrical Engineering, Grove School of Engineering, City College of the City University of New York, New York, NY, USA. [2]Physics Program, Graduate Center of the City University of New York, New York, NY, USA. [3]Department of Physics, City College of New York, New York, NY, USA. [4]Nokia Bell Labs, New Providence, NJ, USA. [5]Advanced Science Research Center, City University of New York, New York, NY, USA. ✉e-mail: akhanikaev@ccny.cuny.edu

direction[37], the property which intimately relates to the structure of Maxwell's equations[38]. Recently this property was leveraged to demonstrate the photonic spin-Hall effect[39,40]. Surprisingly, similar coupling of angular momentum to propagation direction was predicted for evanescent acoustic pressure waves, despite their scalar nature[41,42], leading to some exciting parallels with relativistic physics and topological interpretation of acoustic surface waves[43], acoustics-electromagnetics analogies[44], and unveiled radiation forces and torques produced by acoustic fields[45]. This opens new possibilities to control propagation of acoustic fields through the angular momentum[46–51]. Here we show that in periodic systems the degree of "chirality" of acoustic field, represented by the magnitude of the local transverse angular momentum of the velocity field, can be highly nonuniform and we exploit it to implement directional excitation of acoustic surface waves.

As the first example of a structure with coupled angular and linear momenta we consider a metasurface formed by an array of square holes drilled in a mechanically hard material (high density polyethylene, or HDPE), as shown in Fig. 1a. The holes are isolated from one another by walls in the HDPE material and support the lowest frequency (fundamental) acoustic mode with pressure and velocity fields oscillating in the vertical direction. In the array, the coupling between the modes of the holes takes place only via free space. The arrays of holes have been of significant interest in photonics and were shown to host guided modes, referred to as spoof plasmons. Similar guided modes appear in the case of acoustic hole arrays[48–51], which represent collective acoustic oscillations in the holes that are evanescently coupled through free space. Such acoustic guided modes appear below the sound-line in free-space and therefore are also evanescent in far-field and do not experience radiative loss as they propagate. Figure 1b shows the dispersion of such modes obtained with first-principles calculations in a finite element solver COMSOL Multiphysics, Acoustic Module, . More importantly, they reveal a chiral character of the velocity field, which rotates oppositely for waves propagating in opposite directions. The chirality of the velocity field appears to be uniquely related to the linear Bloch momentum $\mathbf{k} = (k_x, k_y)$ of the surfaces wave, as illustrated by the vector field calculated at $\nu = 5200$ Hz. Here, we define chirality of the acoustic field as the degree of circular polarization of the velocity field evaluated by projecting the normalized velocity field onto right-circularly polarized (RCP) state

vectors in the X-Z (1, 0, $i$) and Y-Z (0, 1, $i$) planes, yielding a two-vector $\mathbf{S}$. Important observation from numerical simulations is that the chirality $\mathbf{S}$, shown in Fig. 1b insets, is highly nonuniform, with hotspots above the holes where the chirality reaches its extreme value of perfect RCP or perfect left-circularly polarization (LCP). Therefore, the directionality of excitation can be maximized by placing a circularly polarized source of the velocity field at the hotspot. The chiral nature of the surfaces wave implies that by selecting the plane of excitation (e.g., xz-plane vs yz-plane) of a circularly polarized source allows controlling the propagation direction of the wave along the surface. The field profiles in Fig. 1c also confirm that the modes are exponentially decaying in the vertical direction and are guided along the surface.

In Fig. 1d we show experimental results for a source producing circularly polarized velocity fields placed at the hotspot (for $\nu = 5600$ Hz). The source is realized with two orthogonal linear acoustic transducers (speakers) oscillating with a relative phase-shift of 90 and −90 degrees to implement right- and left-handed excitations, respectively, in the x–z plane (see Methods for details). As seen in Fig. 1d, the two measurements performed for opposite handedness of the source lead to highly directional excitation of surface waves. We thus confirmed the possibility of controlling the propagation direction and directionality of excitation by local coupling to the hotspot of the transverse angular momentum carried by the evanescent fields of acoustic "spoof plasmons". The nonuniform distribution of chirality also provides the opportunity to enhance such directional excitation, which may be of special importance for practical applications.

## Kagome lattice with spin-polarized bulk states

As a second example, we study a two-dimensional acoustic kagome lattice[52,53], also known to possess a topological phase characterized by nontrivial bulk polarization when its geometry is distorted[22,23]. The unit cell of the lattice is shown schematically in Fig. 2a and it is formed by an array of acoustic resonator trimers coupled via narrow rectangular channels. Each resonator hosts acoustic pressure modes oscillating in the axial direction. We choose to work with the fundamental mode (~5200 Hz), which has its only node at the center of the resonant cavity. The coupling strength between the resonators is tuned by shifting the channels closer or farther away from the center node, thus enabling fine control over the local coupling strength. Due to only nearest

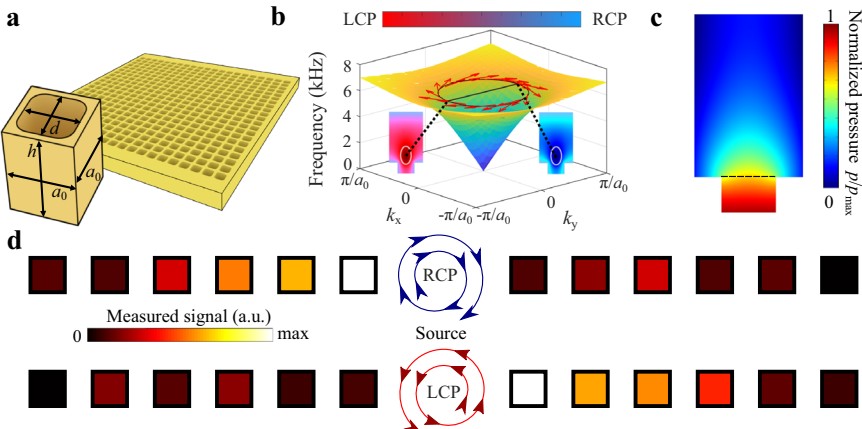

**Fig. 1 | Transverse angular momentum in the near-field and directional excitation of acoustic waves in 2D hole arrays. a** Schematic of the hole unit cell and of the array (pictured behind) drilled in a high-density polyethylene. Dashed line is the high symmetry Γ-X direction chosen for the results of **d**. For the case of (45° tilted) high symmetry Γ-M direction, see Supplemental Fig. 3. **b** Dispersion of guided modes supported by the acoustic hole array with chirality of the velocity field characterized by $\mathbf{S} = (S_{XZ}, S_{YZ})$, where $S_{XZ}$ ($S_{YZ}$) is the chirality components of the evanescent field at the hot spot within the X-Z (Y-Z) plane. Vector field $\mathbf{S}$ in **b** is calculated by projecting the normalized velocity field vector $|\psi\rangle$ at each point in the momentum space onto RCP basis within the respective plane, $\langle RCP_{X-Z(Y-Z)}|, |\psi\rangle$, which yields two chirality components of $\mathbf{S}$. The projection is then rescaled such that LCP (initially 0) maps to −1 and in-phase fields (initially $1/\sqrt{2}$) maps to 0 for a consistent domain of [−1 1]. Inset: simulation of the circularly polarized hotspot, blue representing that the field is purely right-circularly polarized $\mathbf{S} = (0,1)$, red representing pure left-circular polarization $\mathbf{S} = (0,−1)$ at two opposite momenta for wave propagating along x-direction. **c** Pressure field profile of the modes located at the dotted black lines of **b**. **d** Experimental results along the dashed line in **a** demonstrating directional excitation.

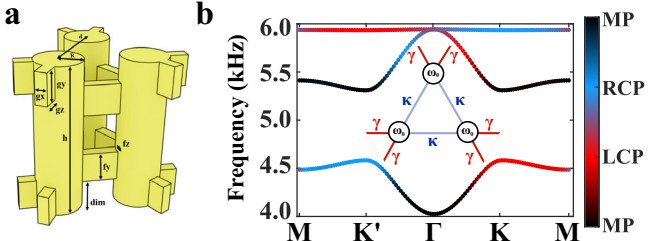

**Fig. 2 | The transition between linear polarization to circular polarization for acoustic modes in kagome lattice with synthetic degrees of freedom. a** Unit cell of the acoustic kagome crystal with in-plane synthetic pseudo-spin defined as circularly polarized rotation of acoustic pressure field within the cell (phase shift of $2\pi/3$ between resonators). **b** First-principles acoustic band structure of the crystal with color indicating the degree of RCP, LCP, and in phase (IP) polarizations. Color bar data is calculated by first building a normalized wavefunction of the pressure amplitude at the antinode of each resonator in the trimer unit cell such that $|u_n(\mathbf{k})\rangle = (P_1, P_2, P_3)$ and then evaluating the symmetry related property $\phi_{C_3} = -i\log(\langle u_n(\mathbf{k})|R_3|u_n(\mathbf{k})\rangle)$.

neighbor connectivity, this system can be mapped onto the tight-binding model (TBM)[22] with the nearest-neighbor coupling described by inter-cell $\gamma$ and intra-cell $\kappa$ coupling parameters (Fig. 2b inset).

The use of a trimer as a unit cell to engineer a synthetic pseudo-spin is justified by the fact that it enables angular momentum through $C_3$ rotational symmetry. The symmetry ensures that the trimer hosts two dipolar modes exhibiting out of phase oscillation in the unit cell in addition to the monopole mode showing uniform phase in the lattice. This feature is inherited by the kagome lattice and it enables the use of the angular momentum within the unit cell to control propagation of collective Bloch states in the array. Indeed, the inspection of the field profiles calculated in COMSOL Multiphysics for different bands confirms this property. We quantify the chirality of the modes based on the $C_3$ symmetry properties of kagome lattice as $\phi_{C_3} = -i\log(\langle u_n(\mathbf{k},\mathbf{r})|R_3|u_n(\mathbf{k},\mathbf{r})\rangle)$, where $u_n(\mathbf{k},\mathbf{r})$ is the eigenstate of n-th band for the unit cell, $\mathbf{k}$ is the Bloch wavevector of the eigenstate,
$$R_3 = \begin{pmatrix} 0 & 1 & 0 \\ 0 & 0 & 1 \\ 1 & 0 & 0 \end{pmatrix}$$
is the three-fold rotational operator in real space[22]. When the chirality, or equivalent $\phi_{C_3} = 0\,\mathrm{mod}.1$, it indicates the eigenmode has uniform phase in space and is monopolar mode. When $\phi_{C_3} = \pm\frac{1}{3}\,\mathrm{mod}.1$, the eigenmodes are exactly circularly polarized with opposite chiralities. When $\phi_{C_3} = 1/2\,\mathrm{mod}.1$, the eigenmode is out of phase in space and belongs to dipolar mode.

The band diagram with $\gamma > \kappa$ (expanded case) in Fig. 2b, obtained using the first-principles finite element method, shows color-coded bands with chirality varying from zero to one in momentum space. For example, the low-frequency band changes its chirality from left-handed (corresponding to $\phi_{C_3} = \frac{1}{3}$, red color) at K point to right-handed ($\phi_{C_3} = -\frac{1}{3}$, blue color) at K' point, to monopolar field profile ($\phi_{C_3} = 0\,\mathrm{mod}.1$, black color) at $\Gamma$ point, and to dipolar field profile ($\phi_{C_3} = 1/2$, purple color) at M point. The second low-frequency band carries almost zero chirality along the high symmetry path between K/K' points and M point but its chirality increases to 1/2 when the momentum approaches the point $\Gamma$. The highest frequency flat band is nearly left- and right-handed circularly polarized modes everywhere except close to high symmetry points $\Gamma$ and M, where its chirality is 1/2, as revealed by Fig. 2b. The frequency bands of positive and negative Bloch momentum are time-reversal partners which transform one onto another after a time-reversal operation, flipping their wavenumber and handedness. Therefore, Fig. 2b clearly shows that the maximal degree of handedness for low-frequency band is achieved at K and K' points where the modes are pure circularly left- and right-

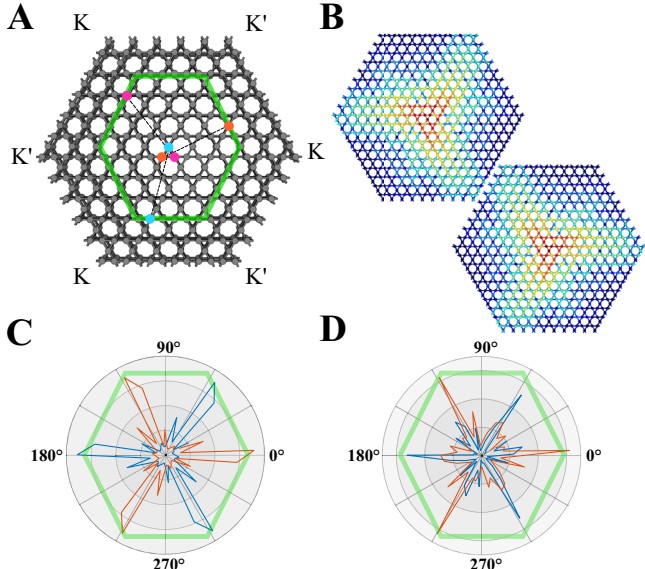

**Fig. 3 | Experimental demonstration of selected acoustic mode propagation. a** Map of the kagome lattice with measured path outlined in green and source located at the central blue site. Each corner of the hexagon represents the K/K' points of the Brillouin zone. When measuring the amplitude and phase response along the green path, we considered the $C_3$ symmetry of the lattice: placing a source at the central blue site and measuring the amplitude/phase response at the blue site on the green path is equivalent to placing the source at the central orange (magenta) site and measuring the amplitude/phase response at the orange (magenta) site along the green path. At each site $n$ along the green path we can now describe the full amplitude response for a circularly polarized source as a superposition of the contribution from each of the individual source sites: $\widetilde{A}_n = \widetilde{A}_{blue,n} + \widetilde{A}_{orange,n} e^{\pm i(2\pi/3)} + \widetilde{A}_{magenta,n} e^{\pm i(4\pi/3)}$. **b** Large scale first-principles calculation of sources of different handedness placed at the center. **c** Angular distribution of the numerically calculated amplitude response. **d** Experimentally measured angular distribution of the full amplitude response $\widetilde{A}_n$. Orange represents left-circularly polarized (LCP) superpositions and blue represents right-circularly polarized (RCP) superpositions. LCP sources propagate along the K direction while RCP sources propagate along the K' direction.

handed modes. Notably, these are the high-symmetry points where the Dirac points observed for undimerized ($\gamma = \kappa$) lattice.

Importantly, the handedness of rotation of the dipole modes is reversed as the propagation direction is flipped, indicating that handedness can be used to directionally excite Bloch modes of the kagome lattice at K and K' valleys. To demonstrate directional excitation due to the coupling of angular and Bloch momenta, the system is set up according to Fig. 3a, with K and K' direction indicated. When a left- or right-handed circularly polarized source, rotation parallel to the plane of the structure, is placed in the bulk of the crystal, large scale first-principles calculations indeed predict that the modes propagate only along the respective $\Gamma$-K or $\Gamma$-K' directions, as shown in Fig. 3b. The numerical and experimentally measured directionality diagrams for circularly polarized sources of opposite handedness placed in the center of the 3D printed kagome lattice of the same size as in Fig. 3a are plotted in Fig. 3c, d. These results clearly show that the reversal of handedness of the source leads to highly directional excitation only along the respective direction in the Brillouin zone, with left-handed (right-handed) source exciting modes propagation along K (K') direction. We note that the directionality takes place for wide range of values of Bloch momenta, not only at high-symmetry points. The use of synthetic degrees of freedom originating from the lattice symmetries thus opens another powerful opportunity for controlling the directionality of excitation in acoustic systems. We note that this behavior would also occur for undimerized and $\gamma < \kappa$ cases and isn't related to the bulk topological properties of the kagome lattice.

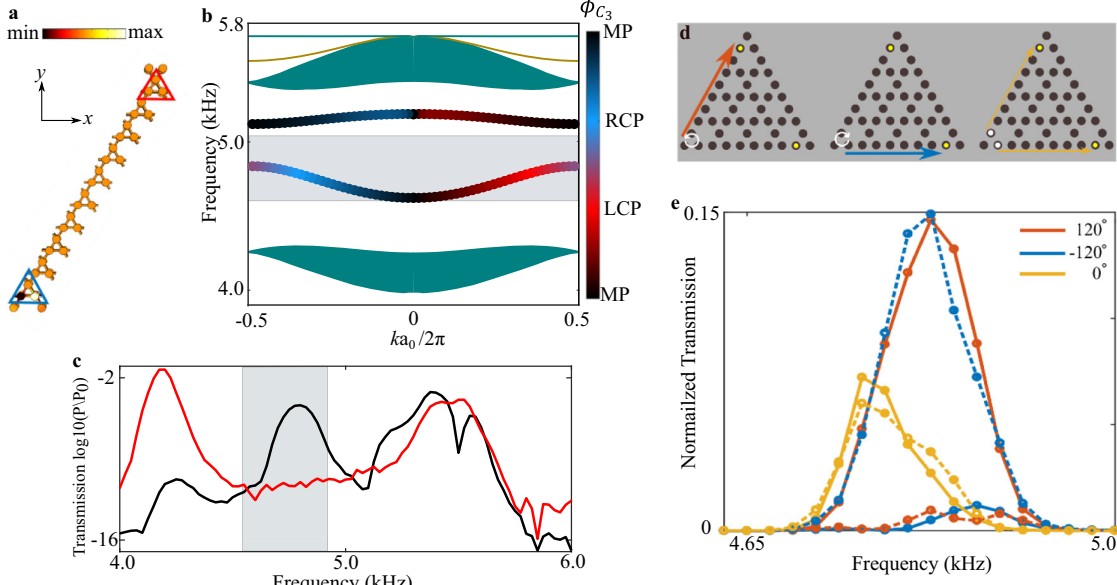

**Fig. 4 | Experimental demonstration of angular momentum (pseudo-spin) locking of the edge states. a** Field distribution at the lower edge band in a supercell consisting of 10 cells near $k_x a_0 = \pi$, the supercell is periodic in the $x$ direction and terminated by a boundary cell in the $y$ direction. **b** Band structures of a supercell terminated by cylinders with half height of the trimers. The dark yellow line denotes edge states located outside the bulk bandgap, and teal shaded regions are the bands of bulk states. The color encoded in the edge bands represent normalized $\phi_{C_3}$ of the lower (blue) and upper (red) edge band, calculated for the edge states of the supercell, denoted by blue and red triangle in **a**, respectively.

**c** Transmission spectra for the expanded lattice (black line) and shrunken lattice (red line), respectively. The shaded light gray regions in **b** and **c** indicate the frequency region of the edge state of interest. **d** Schematics of edge state excitation by different sources; the sources (from left to right) are left-handed circular polarized, right-handed circular polarized and linearly polarized. Yellow dots are the probing positions and white dots are the source position. **e** Normalized transmission spectra measured at two edges illustrated in **d**, for phase differences between the two sources of 120°, −120°, and 0°, respectively. Solid and dashed lines are the spectra measured at the upper left and bottom right yellow dots in **d**.

## Directional emission of topological edge states in acoustic kagome lattice

As the symmetry is broken by detuning inter-cell $\gamma$ and intra-cell $\kappa$ couplings between neighboring trimers, a topological transition takes place in the kagome lattice[52]. Specifically, the trimer in Fig. 2a support bandgaps at both K and K′ valleys with topologically nontrivial ($\gamma > \kappa$) and trivial ($\gamma < \kappa$) nature (referred to as expanded and shrunken respectively), implying that a control over the coupling parameters can enable ad-hoc topological transitions.

We note that kagome lattices are sometimes considered as valley-Hall insulator[54,55] which is rather superficial treatment, because valley-Chern number is not a good topological invariant and Wannier bulk polarization should be used instead. Although the effects of spin-momentum locking reported here can be found in other topological systems, including true valley-Hall systems[32,33] the difference is clearly seen in the strongly gapped nature of the edge states, which indicates that the respective bulk-boundary correspondence, that would manifest in valley-Hall systems in the form of gapless edge states, does not hold for kagome latices.

Topological edge states due to the nonvanishing bulk polarization in the topological domain emerge at the boundary between topologically distinct crystals, and they have remarkable properties associated with their topological nature. In particular, the robustness of topological states in TR-symmetric systems has been attributed to spin-momentum locking in spin-Hall effect type systems[7,56–58]. Even though the system presented here belongs to a completely different topological class[52], we found a similar locking associated to angular momentum (pseudo-spin) in the edge states of this system. This property stems from the chiral structure of the bulk modes mentioned above. Thus, due to the circularly polarized character of the bulk bands, which are circular right-handed along K direction and circular left-handed along K′ direction, the edge states also exhibit pseudo-spin polarization which reverses for opposite wavenumbers. To

corroborate this argument, we conducted first-principles calculations of the supercell in Fig. 4a. The calculated band structure reveals two types of edge states, localized at the lower end (blue band) and upper end (red band) of the strip (Fig. 4b).

The chirality $\phi_{C_3}$ of the edge states, which is defined for the edge eigenstate distributions over the corresponding edge trimers, varies as function of projected momentum vector $k_x$ and is encoded in the color of the edge bands in Fig. 4b. Though the edge band with higher frequency barely exhibits a circular polarized behavior, the lower edge band does behave like a circularly polarized mode when $k_x$ approaches the projected valley points of the Brillouin zone ($k_x a_0/2\pi = \pm 1/3$).

To experimentally test this idea, the 3D printed triangular-shaped lattice was fabricated with the same boundaries supporting edge states carrying angular momentum at all three boundaries and, therefore, it only holds the edge states of interest, lying in the lower edge band in Fig. 4b. Using the fact that there is one-to-one correspondence among frequency, momentum, and chirality $\phi_{C_3}$, we performed a frequency sweep over the lower edge band (shown as the light gray shaded region in Fig. 4b) with sources carrying different angular momenta. Indeed, due to pseudo-spin locking, the edge modes are expected to be excited unidirectionally, traveling in different directions as a function of the source handedness. As experimentally demonstrated below, this feature enables robust directional excitation of topological edge states.

We next experimentally investigate the nature of these chiral edge states. To this aim we measured the transmission along the edge by placing the source at the corner of both topological and trivial structures, and measured the signal by placing a detector at the center of the opposite edge. The measurement results for both trivial and nontrivial structures are shown in Fig. 4c, clearly revealing the presence of a bandgap in both cases and an additional midgap transmission peak associated with the excitation of edge states in the

topological lattice. The measured frequency range of edge states is consistent with our theoretical predictions (shaded gray region).

To further explore the properties of these acoustic edge modes and exploit their angular momentum to momentum locking, we placed two sources with controllable phase shift at the two sites of the same corner trimer, as shown in Fig. 4d, and measured the amplitude response as the frequency was swept over the range of the blue edge band at the upper left edge and the bottom right edge of the structure (yellow dots in Fig. 4d). We found that the transmissions for the two sites became strongly dependent on the relative phase in the frequency range of edge states. When the phase difference is 120°, representing a left-handed circularly polarized source, stronger excitation occurs at the upper left edge compared to the excitation at the bottom right edge, as shown by red curves in Fig. 4e. Meanwhile, the maximum excitation is observed near the upper end of the edge band (~4.85 kHz), implying that the edge mode with left-handed circular polarization near the projected K-point is excited. Repeating the same measurement when the phase difference is −120°, representing a right-handed circularly polarized source, stronger excitation occurs at the bottom right site (blue curves in Fig. 4e), confirming the rotating character of the modes and the locking of angular momentum to propagation direction. Interestingly, when the phase difference is 0°, the transmissions at the two sites are detected with nearly same magnitude, with a lower maximum value compared to the ones of circular polarized sources, indicating that the energy flux splits equally between the edges. In this case the transmission is maximum at the lower frequency, and it is suppressed at the higher frequency because of the non-circular polarization of the source, consistent with our theoretical prediction in Fig. 4b.

## Discussion

Novel manufacturing techniques such as 3D printing and precise machining which are used to fabricate complex acoustic metamaterials recently allowed researchers to explore a plethora of exciting properties of sound waves. Here, by exploiting two possibilities to control sound in artificial acoustic media, via angular momentum originating from (i) their respective near-field properties and (ii) lattice symmetries, we demonstrated that one can obtain unparalleled capabilities in controlling the wave-propagation in acoustic metamaterials systems. Interestingly, neither of the demonstrated possibilities rely on a spin-Hall topological phase, which is broadly used to achieve spin-polarized transport. For the first approach used here, the elliptically polarized velocity field is a natural consequence of the evanescent nearfield located above an open-ended cavity resonator, where the two degrees of freedom of the elliptical handedness of velocity field effectively model an acoustic pseudospin. For the second approach, the pseudospin is generated entirely through the lattice symmetry, which gives rise to circularly polarized eigenstates for finite linear Bloch momentum. Finally, we demonstrate that in topological systems, one can further expand control over localization of acoustic fields by trapping sound at topological interfaces via the same synthetic pseudo-spin. Thus, synthetic degrees of freedom stemming from engineered properties of sounds waves can provide additional and fascinating ways to directionally excite, filter, and route sound energy, which can be of immense importance for practical applications.

## Methods
### Structure design and generic measurements
The acoustic hole array was fabricated using the desktop CNC Nomad 3 in 5/8 inch HDPE with the design shown in Fig. 1a and dimensions $d = 8$ mm, $h = 10$ mm, and lattice constant $a_0 = 10$ mm. To emulate the circularly polarized source, the speaker was placed 2 mm above the hole at the circular polarization hot spot and two measurements were run: the first when the speaker's normal was parallel to the HDPE's surface

and the second when the speaker's normal was perpendicular to the HDPE's surface. Similarly to the kagome directional excitation measurement (see below), we extracted amplitude and phase data for each hole site and superimposed them onto each other for each orientation of the speaker for a total complex amplitude at each $n$-th site of $\widetilde{A}_n = \widetilde{A}_{\parallel n} \pm i\widetilde{A}_{\perp n}$, addition for emulating a counter-clockwise (CCW) source and subtraction for emulating a clockwise (CW) source. The subscript refers to the orientation of the speaker. The full field profile is plotted in Fig. 1d. The simulation of the polarization of the acoustic velocity field was accomplished in COMSOL Multiphysics FEM acoustic pressure wave modeling for the same parameters as above.

The unit cell design of the kagome lattice is plotted in Fig. 2a. The lattice constant of the structure is $a_0 = 2d = 42.63$ mm, and the height of cylinder is chosen as $h = 40.00$ mm, with the radius $R = 6.20$ mm, such that the frequencies of the desired mode are in the probing range of the microphone. The connectors between the cylinders consist of blocks, with their dimensional sizes $g_x = f_z = 3.01$ mm, $g_y = f_y = 8.00$ mm, and $g_z = 4.47$ mm. The coupling strength of the modes is maximal when the connectors are at the top or bottom of the cylinders, and minimum at the center of the cylinders. In order to make the intra-cell and inter-cell coupling of the trimers inequivalent, the outer connectors of the topological trimer shown in Fig. 2a are placed at the top and bottom of the cylinders, while the inner connectors are shifted toward the center by the distance $dim = 6.00$ mm.

The trimers were fabricated using the B9Creator v1.2 3D printer. All cells were made with acrylic-based light-activated resin, a type of plastic that hardens when exposed to UV light. Each cell was printed with a sufficient thickness to ensure a hard wall boundary condition and narrow probe channels were intentionally introduced on top and bottom sides of each of the cylinders to excite and measure local pressure field at each site. The diameter of the port is $D_0 = 3.73$ mm, and the upper port has a height $H_d = 3.97$ mm, while the height of lower port is $2H_d$. When not in use, the probe channels were closed with 3D printed cups. Each trimer and boundary cells were printed one at a time and the models were designed specifically to interlock tightly with each other. For all measurements, a frequency generator and FFT spectrum analyzer scripted in LabVIEW were used. The FFT spectrum analyzer is also capable of extracting phase differences between two channels.

### Directional excitation measurement
Two speaker were used to excite local pressure fields with controllable relative phase shift (arbitrary waveform generator Rigol DG822) and signal was collected with directional microphones (EMM-6) connected to an external digital data acquisition device (AUBIO BOX USB 96), enabling excitation of the desirable directional propagation. In array of holes structure in Fig. 1, the speakers were placed symmetrically above the hole near the chiral hotspot. For kagome lattice the scheme for emulating a circularly polarized source is shown in Fig. 3a where the speaker is placed at a site in the center of the structure and we record the amplitude and phase response of each site along the green hexagonal path. The phase response is with respect to the source site such that the complex amplitude response for a single measurement is $Ae^{i(\phi_{microphone} - \phi_{source})}$, or the phase difference between the measured sound pressure and the source sound pressure. Then, as stated in the Fig. 3 caption, the complete complex amplitude for any site along the green hexagonal path for a circularly polarized source is the superposition of each of the three complex amplitude measurements from the orange/cyan/magenta source sites multiplied by the respective phase shift. The radial diagram of the full directional response is plotted in Fig. 3c, d for a CCW and CW source, respectively. The simulation in Fig. 3b is accomplished with COMSOL Multiphysics FEM acoustic pressure wave modeling for a CCW and CW source for a substantially large acoustic lattice. We allowed for vast loss in the simulation such that there would be no reflections at the boundary.

## Data availability
Authors can confirm that all relevant data are included in the paper and/or its supplementary information files.

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

## Acknowledgements

The work was supported by the Office of Naval Research (ONR) award N00014-21-1-2092, the National Science Foundation (NSF) grant DMR-1809915, and the Simons Collaboration on Extreme Wave Phenomena. We acknowledge Boris Khanikaev for his assistance in fabricating structures and in setting-up phase-controlled measurements in Fig. 1.

## Author contributions

A.B.K. conceived the idea of the work. M.W. and X.N. performed theoretical studies. M.W. and X.N. performed the numerical design of the structures and carried out experiments. All the authors contributed to the manuscript preparation. A.B.K. and A.A. supervised the research.

## Competing interests

The authors declare no competing interests.

## Additional information

**Correspondence and requests** for materials should be addressed to Alexander B. Khanikaev.

