## [Peer review file · Nature Communications]

REVIEWER COMMENTS

Reviewer #1 (Remarks to the Author):

Referee Report for the manuscript: "Synthetic Pseudo-Spin-Hall Effect in Acoustic Metamaterials",
by Professor Khanikaev and colleagues.

Synthetic pseudo-spin is currently a hot topic in acoustics as they endow scalar pressure waves with additional degrees of freedom, which brings the physics of acoustic systems to the next level of complexity. In this work by Khanikaev's group, the authors introduce several approaches to engineer such additional degrees of freedom and manipulate acoustic pressure waves by their pseudo-spins of two different nature. In the first case, the authors demonstrate that the nearfield of acoustic structure can be highly non-uniformly polarized, with strong "hotspots" of circularly polarized components.

The authors demonstrate that these hotspots are perfect locations to directionally excite the surface waves. In the second case, they show that the rotational C3 symmetry of the kagome acoustic crystal leads to another kind of in-place pseudo spin, which too is leveraged to directionally excite modes along the edges and in the bulk of the 3D printed kagome crystal. While spin-polarized transport has been demonstrated before for edge states in different types of systems (e.g., in spin-Hall type of topological insulator mentioned by the authors), the bulk directionality is shown for the first time to the best of my knowledge. From my point of view, this work is an excellent example of how engineered pseudo-spins can be used to control scalar waves and it reports an important advance for the field of acoustic.

Summarizing everything said above, and also considering a very high likelihood that this paper will be highly cited, I recommend this work for publication after the authors address several minor comments given below.

1. The list of references is incomplete as synthetic degrees of freedom, similar to the pseudo-spin, were used in other contexts in acoustics. Very recent relevant works include the following: Bliokh et al., Klein-Gordon Representation of Acoustic Waves and the Topological Origin of Surface Acoustic Modes, PRL (2019); Toftul, et al., Acoustic Radiation Force and Torque on Small Particles as Measures of the Canonical Momentum and Spin Densities, PRL (2019). Leykam, et al., Edge modes in two-dimensional electromagnetic slab waveguides: Analogs of acoustic plasmons, PRB (2020). These are all very recent and relevant.

2) I consider it as a great advance that the effect of angular-to-linear momentum locking due to the evanescent wave is shown for a 2D system. All prior results were for 1D systems (e.g. of very narrow slits). Therefore, I believe it would be good to emphasize not only the nonuniformity of the field, but also show the spin locking in both directions, at least theoretically. The most effective way of doing this would be to show surfaces of eigenfrequencies instead of a line along one high-symmetry direction in Fig. 1B. The chirality then can be shown as a spatially average and encoded somehow in this surface (similar to Fig. 2B).

3) In the sentence “When the intensity is zero, the mode is full linearly polarized; when it’s one, the mode is full circularly polarized” it is not clear what “one” is. The figure is colored with red and blue which correspond to the LCP and RCP components of the modes, but no “intensity” values are given. Also, one would expect -1, and not 1 for the RCP case. I believe this point should be clarified and the sentence rewritten.

4) The authors use work “mimicking” in two sentences describing directional excitation of the edge states. For instance “mimicking a right-handed circularly polarized source”. I believe their combination of two sources can be exactly decomposed into circularly polarized components. Please revise this section.

5) Why not encode the degree of LCP or RCP in Fig. 4 the same way as in Fig. 2 (in panel Fig. 4B and drop panel Fig. 4C)? While the picture does explain the point of momentum locking, this would give more uniform style across the results in the paper. Panel Fig. 4C then could be moved to the supplement if authors prefer to have it somewhere in the manuscript.

I would like to provide some additional comments to improve the presentation of this otherwise excellent work:

*) several of the figures look a bit blurred and fuzzy, instead of crisp and sharp. Using INKSCAPE or similar software would allow making VECTOR figures, which would be sharper and crisper.

*) the vertical axis of Figure 1B has several obvious problems (when reading the printed version of the manuscript. Perhaps these problems are less severe for readers who have very large monitors, but some people read papers in small laptops).

*) The vertical axis of Figure 1 B has 24 zeros. There is no need to print two dozen zeros. If the units are chosen as kHz, instead of Hz, then two dozen zeros would not be needed. Also, there is no need to write so many numbers there. 1, 3, 5, and 7 seem to be more than enough. The plots is quite simple.

*) The horizontal axis of Figure 1 B has two numbers: 0 and 1. These two numbers are enough. On the vertical axis, it seems that 1 and 7 should be enough. Just two numbers on each axes. The vertical one could have more, if desired, like 1, 3, 5, and 7. But not dozens of unnecessary numbers there.

*) The vertical axis of Figure 1B has a label [Frequency (Hz)] which is way too small. In my printed version, I need a magnifying glass. Also, it looks a bit blurred and fuzzy, instead of crisp and sharp.

*) The horizontal axis of Figure 1B has a label which is way too small. In my printed version, I need a magnifying glass. Also, it looks a bit blurred and fuzzy, instead of crisp and sharp.

*) The figure caption of fig. 1 writes $f = ck/2\pi$ but the math symbols I think should be in italics.

*) Figure 2 B has three black circles at the center of the figure and (in the printed version) I cannot read these well. Are these ω_0 ? The subindex is very fuzzy and blurred. Again, using VECTOR figures would help. But better not to use nano-scale font sizes. Better to use larger font sizes in many parts of ALL the figures.

*) Figure 2 B has six ghostly gammas. Very faint. Many data points are almost invisible, very faint, quite ghostly, especially in the upper band.

*) Figure 3 caption, the subindices: blue, orange, magenta, should be in roman font, not italics.

*) Fig. 4: the vertical axes have text labels which are too small, too blurred, and fuzzy. And way to many unnecessary numbers (especially panels B,C,D)

*) through the paper, better to add a space before units. And the units "mm" should be in roman font.

Overall, this is an excellent work in a very interesting area of research, which is attracting growing attention. The contents are sufficiently important to be published in Nature Comm. and I think that this work will attract considerable attention from the research community.

Reviewer #2 (Remarks to the Author):

In the manuscript entitled "Synthetic pseudo-spin-Hall effect in acoustic metamaterials" by Matthew Weiner et al., the authors proposed two kinds of acoustic metamaterials systems, which support the called "synthetic" pseudospin-related sound wave propagation. The first one is built based on the perforated plate benefitted from the evanescent waves, while the other one is constructed by the Kagome acoustic lattice. Although the demonstrations are reported in both simulations and experiments, the reviewer really feels the reported results in this work are not sufficient for publication in this high-impact journal.

This work is not innovative enough and some of the reported results have been studied to the best of my knowledge. In the first scenario, the proposed acoustic analogous spin-Hall effect based on the drilled cavity structures has been demonstrated in Refs. [44] and [45]. In the second scenario, the proposed chirality-based bulk mode propagation (Fig. 3) and directional edge emission (Fig. 4) have been well studied in previous works about acoustic valley-Hall phases and can be clearly explained by the valley-projected physics, such as valley pseudospin and valley-Hall topological edge states in Refs. [PRB 95, 174106 (2017); Nat. Phys. 13, 369 (2017); PRL 120, 246601 (2018)].

There are other technical issues that need to be addressed.

1. In Fig. 1D, only one-dimensional spatial field distributions is provided. Sound propagations along the other direction in the x-y plane should also be discussed here to confirm the unidirectional excitation.

2. Band diagram for the real acoustic lattice should be added into Fig. 2B to compare with the TBM results.

3. During the measurements of the field distributions by using the chiral source, the authors describe the full amplitude response as a superposition of the contribution from each of the individual source sites. Why not use the several sources simultaneously? Is it a technical problem or is there any difference between these two methods please?

4. Please have a double check on the writing, which includes some typos and misleading:

- 4.1) What do CCW and CW stand for?

4.2) Line 81: "...field. and this..." -> "...field. And this...".

4.3) A red dot is missed in Fig. 1B.

4.4) What is Jones vector?

4.5) Line 179: "G-K" -> " Γ -K".

Reviewer #3 (Remarks to the Author):

In the manuscript by Weiner et al., titled "Synthetic Pseudo-Spin-Hall Effect in Acoustic Metamaterials", the authors investigate two types of synthetic pseudospins, which emerge either due to the evanescent nature of modes or due to symmetry of the lattice. While the first type of pseudospin was reported before, the authors bring novelty by considering inhomogeneous character of the nearfield in patterned metasurfaces. This feature is absent in modes localized due to excitation of internal degrees of freedom of materials (e.g., plasmons in metal) and in deep subwavelength limit considered so far. Therefore, I consider this result as a significant step forward, especially because the authors directly demonstrate that this property can be used for highly efficient directional excitation by placing circularly polarized sources at the chiral hotspots of the near-field. The second section, which focuses on the symmetry-engineered pseudo-spin, is also absolutely new to the best of my knowledge. While Kagome structures have been used to demonstrate topological properties and higher-order topological (corner) states by the authors, here they focus on a different aspect which largely evaded attention of the acoustic community. Thus, this work shows that both bulk and edge states can be directionally excited in both 2D (across the surface of the structure) and in 1D (along the edges) with very high efficiency.

The paper also very well-written and makes strong case for the use of synthetic pseudospins in acoustic metamaterials, which is likely to attract a significant interest from the Nature Communications readership. I there recommend this work for publication, after a minor correction.

1) One small correction needed. In Fig 1C, LP should be replaced with LCP.

2) I also would like to ask for a clarification. From my point of view the system does not have to be topological to exhibit spin-Hall effect in the bulk. It would be good to clarify this point in the revision.

AUTHORS' RESPONSE TO THE REVIEWERS' COMMENTS

Reviewer #1:

General remarks by Reviewer #1

Synthetic pseudo-spin is currently a hot topic in acoustics as they endow scalar pressure waves with additional degrees of freedom, which brings the physics of acoustic systems to the next level of complexity. In this work by Khanikaev's group, the authors introduce several approaches to engineer such additional degrees of freedom and manipulate acoustic pressure waves by their pseudo-spins of two different nature. In the first case, the authors demonstrate that the nearfield of acoustic structure can be highly non-uniformly polarized, with strong "hotspots" of circularly polarized components.

The authors demonstrate that these hotspots are perfect locations to directionally excite the surface waves. In the second case, they show that the rotational C_3 symmetry of the kagome acoustic crystal leads to another kind of in-place pseudo spin, which too is leveraged to directionally excite modes along the edges and in the bulk of the 3D printed kagome crystal. While spin-polarized transport has been demonstrated before for edge states in different types of systems (e.g., in spin-Hall type of topological insulator mentioned by the authors), the bulk directionality is shown for the first time to the best of my knowledge. From my point of view, this work is an excellent example of how engineered pseudo-spins can be used to control scalar waves and it reports an important advance for the field of acoustic.

Summarizing everything said above, and also considering a very high likelihood that this paper will be highly cited, I recommend this work for publication after the authors address several minor comments given below.

Authors' response to general remarks by Reviewer #1.

We thank the reviewer for carefully reading our manuscript and for providing their positive comments and valuable suggestions. We have addressed Reviewer #1 comments accordingly in the following reply and we believe addressing these points have improved the manuscript. The suggestions regarding revisions to the figures, to emphasize the 2D nature of our systems were particularly useful, as they clearly allowed us to showcase the novelty and originality of our results and contrast them with earlier studies. We also thank Reviewer #1 for noting that our work will likely attract a significant attention and will be highly cited.

Reviewer #1 specific remark 1

1. The list of references is incomplete as synthetic degrees of freedom, similar to the pseudo-spin, were used in other contexts in acoustics. Very recent relevant works include the following: Bliokh et al., Klein-Gordon Representation of Acoustic Waves and the Topological Origin of Surface Acoustic Modes, PRL (2019); Toftul, et al., Acoustic Radiation Force and Torque on

Small Particles as Measures of the Canonical Momentum and Spin Densities, PRL (2019).
Leykam, et al., Edge modes in two-dimensional electromagnetic slab waveguides: Analogs of acoustic plasmons, PRB (2020). These are all very recent and relevant.

Authors' response to specific remark 1

We admit that we overlooked the works suggested by Reviewer #1, which indeed are very relevant. We have added the respective references in the proper context in the revised manuscript.

Reviewer #1, specific remark 2

I consider it as a great advance that the effect of angular-to-linear momentum locking due to the evanescent wave is shown for a 2D system. All prior results were for 1D systems (e.g., of very narrow slits). Therefore, I believe it would be good to emphasize not only the nonuniformity of the field, but also show the spin looking in both directions, at least theoretically. The most effective way of doing this would be to show surfaces of eigenfrequencies instead of a line along one high-symmetry direction in Fig. 1B. The chirality then can be shown as a spatially average and encoded somehow in this surface (similar to Fig. 2B).

Authors' response to the specific remark 2

We thank Reviewer #1 for this very useful suggestion. Following Reviewer #1 recommendation, we have revised Figure 1B by plotting the 2-dimensional energy bands of the structure, instead of the linear dispersion of the previous version of the manuscript. We have also presented the “helicity” and spin-momentum looking of the modes by calculating projections of the velocity field vorticity/chirality both in the X-Z and Y-Z planes, and across the entire constant frequency contour. To do so, we construct the pseudo-spin vector field $\mathbf{S}=(S_x, S_y)$ where S_x is chirality in the Y-Z plane and S_y is chirality in the X-Z plane, defined as projections of the velocity field on circularly polarized basis in the respective planes. This form of presentation is indeed best suited to emphasize the 2D nature of our system.

Reviewer #1, specific remark 3

In the sentence “When the intensity is zero, the mode is full linearly polarized; when it's one, the mode is full circularly polarized” it is not clear what “one” is. The figure is colored with red and blue which correspond to the LCP and RCP components of the modes, but no “intensity” values are given. Also, one would expect -1, and not 1 for the RCP case. I believe this point should be clarified and the sentence rewritten.

Authors' response to the specific remark 3

We thank the Reviewer for this helpful suggestion. We used two quantities to describe the polarization of the modes in the old manuscript: the color of the line represents the degree of polarization between LCP and RCP, and the opacity of the line encodes the degree of monopolar mode. To avoid confusion, in the revised manuscript, we use the approach employed in Fig.4 to quantify the chirality of the modes based on the C_3 symmetry properties of the kagome lattice. The formula for chirality is then written as $\phi_{C_3} = -i \log (\langle u_n(k, r) | R_3 | u_n(k, r) \rangle)$, where $u_n(k, r)$ is the eigenstate of n^{th} band for the unit cell, k is the wavevector of the eigenstate, $R_3 = \begin{pmatrix} 0 & 1 & 0 \\ 0 & 0 & 1 \\ 1 & 0 & 0 \end{pmatrix}$ is the three-fold rotational operator in real space. When the chirality, or equivalent $\phi_{C_3} = 0 \text{ mod. } 1$, it indicates the eigenmode has uniform phase in space and is a pure monopolar mode. When $\phi_{C_3} = \pm \frac{1}{3} \text{ mod. } 1$, the eigenmodes are exactly circularly polarized with opposite chiralities. When $\phi_{C_3} = 1/2 \text{ mod. } 1$, the eigenmode is out of phase in space and represents a dipolar mode. Any other polarized states in Kagome lattice can be linearly constructed based on the listed cases. Therefore, the quantity ϕ_{C_3} , modulus in 1, is solely used to describe the chirality of all polarized modes in Kagome lattice.

Reviewer #1, specific remark 4

The authors use word “mimicking” in two sentences describing directional excitation of the edge states. For instance “mimicking a right-handed circularly polarized source”. I believe their combination of two sources can be exactly decomposed into circularly polarized components. Please revise this section.

Authors' response to the specific remark 4

We thank Reviewer #1 for noticing this awkward wording. We have revised changed the word “mimicking” to “representing”.

Reviewer #1, specific remark 5

Why not encode the degree of LCP or RCP in Fig. 4 the same way as in Fig. 2 (in panel Fig. 4B and drop panel Fig. 4C)? While the picture does explain the point of momentum locking, this would give more uniform style across the results in the paper. Panel Fig. 4C then could be moved to the supplement if authors prefer to have it somewhere in the manuscript.

Authors' response to the specific remark 5

We thank Reviewer #1 for this useful suggestion, and we have revised the figure accordingly. Thus, we have encoded the degree of LCP/RCP (represented by ϕ_{C_3} , as explained in detail in the

reply to the comment (3)) in the band structures of Fig.4B which matches the style in Fig. 2 and drop Fig.4c.

Reviewer #1, minor/technical remarks and suggestions

- *) several of the figures look a bit blurred and fuzzy, instead of crisp and sharp. Using INKSCAPE or similar software would allow making VECTOR figures, which would be sharper and crisper.
- *) the vertical axis of Figure 1B has several obvious problems (when reading the printed version of the manuscript. Perhaps these problems are less severe for readers who have very large monitors, but some people read papers in small laptops).
- *) The vertical axis of Figure 1 B has 24 zeros. There is no need to print two dozen zeros. If the units are chosen as kHz, instead of Hz, then two dozen zeros would not be needed. Also, there is no need to write so many numbers there. 1, 3, 5, and 7 seem to be more than enough. The plots is quite simple.
- *) The horizontal axis of Figure 1 B has two numbers: 0 and 1. These two numbers are enough. On the vertical axis, it seems that 1 and 7 should be enough. Just two numbers on each axes. The vertical one could have more, if desired, like 1, 3, 5, and 7. But not dozens of unnecessary numbers there.
- *) The vertical axis of Figure 1B has a label [Frequency (Hz)] which is way too small. In my printed version, I need a magnifying glass. Also, it looks a bit blurred and fuzzy, instead of crisp and sharp.
- *) The horizontal axis of Figure 1B has a label which is way too small. In my printed version, I need a magnifying glass. Also, it looks a bit blurred and fuzzy, instead of crisp and sharp.
- *) The figure caption of fig. 1 writes $f = ck/2\pi$ but the math symbols I think should be in italics.
- *) Figure 2 B has three black circles at the center of the figure and (in the printed version) I cannot read these well. Are these ω_0 ? The subindex is very fuzzy and blurred. Again, using VECTOR figures would help. But better not to use nano-scale font sizes. Better to use larger font sizes in many parts of ALL the figures.
- *) Figure 2 B has six ghostly gammas. Very faint. Many data points are almost invisible, very faint, quite ghostly, especially in the upper band.
- *) Figure 3 caption, the subindices: blue, orange, magenta, should be in roman font, not italics.
- *) Fig. 4: the vertical axes have text labels which are too small, too blurred, and fuzzy. And way to many unnecessary numbers (especially panels B,C,D)
- *) through the paper, better to add a space before units. And the units “mm” should be in roman font.

Authors' response to the minor/technical remarks and suggestions

We are grateful to Reviewer #1 for their thorough reading and for paying attention to the details. We have addressed all the issues noted by the reviewer and we have reformatted the paper by embedding high-resolution (600 DPI) figures generated from the Inkscape vector source files.

Reviewer #1, concluding remark

Overall, this is an excellent work in a very interesting area of research, which is attracting growing attention. The contents are sufficiently important to be published in Nature Comm. and I think that this work will attract considerable attention from the research community.

Authors' response to the reviewer's concluding remarks

We thank once again Reviewer #1 for his/her highly useful remarks, which, we believe, have allowed us to significantly improve the manuscript, to emphasize novelty, and to make our results more accessible to broader readership. We very much appreciate the positive Reviewer's feedback and his/her strong recommendation to publish our work in Nature Communications.

Reviewer #2:

General remarks by Reviewer #2

In the manuscript entitled "Synthetic pseudo-spin-Hall effect in acoustic metamaterials" by Matthew Weiner et al., the authors proposed two kinds of acoustic metamaterials systems, which support the called "synthetic" pseudospin-related sound wave propagation. The first one is built based on the perforated plate benefitted from the evanescent waves, while the other one is constructed by the Kagome acoustic lattice. Although the demonstrations are reported in both simulations and experiments, the reviewer really feels the reported results in this work are not sufficient for publication in this high-impact journal.

This work is not innovative enough and some of the reported results have been studied to the best of my knowledge.

Authors' response to the general remarks

First of all, we would like to thank Reviewer #2 for sharing their opinion about our work and for their remarks. We find these remarks useful for improving the manuscript, yet we kindly disagree with the comments about the novelty of our work. As detailed below, there are several aspects which put our work apart from any earlier works, and in the revise version of the manuscript we farther emphasized these novelty aspects. In what follows, we provide our detailed response to Reviewer #2 remarks and suggestions.

Reviewer #2, specific remark 1

In the first scenario, the proposed acoustic analogous spin-Hall effect based on the drilled cavity structures has been demonstrated in Refs. [44] and [45].

Authors' response to the specific remark 1

We thank Reviewer #2 for this remark, as it shows that there is some misunderstanding on the Reviewer's side about our work. Regarding "the first scenario", the first major difference of our

results as compared to “spin-locking” in pioneering work [44], is that we report spin-locking in 2D system and not in 1D system. Indeed, Ref. [44] reports the chiral nature of the evanescent field and the spin-momentum locking properties, but does it for the array of grooves periodically arranged in one dimension, while our system is two dimensional. This could have evaded Reviewer #2 attention as the plot spin-locking in previous version of the paper was somewhat confusing and plotted only one high symmetry direction. However, by the request from Reviewer #1, we replotted Fig. 1 to reflect 2D nature of our system and the fact that spin-momentum takes place for propagation in any direction along the metasurface. This property is in fact of immense importance and sets our work apart from any earlier works as we can directionally excite modes in any direction by using the spin-locking. Second major difference lays in the fact that Ref. [44] focuses on the subwavelength regime when the field is highly “homogenized” and yields a relation between velocity field components in the form “ $v_x = \pm ikv_y/\tau$ ” (expression from Ref. [44]). As can be seen from this relation, in addition to its clearly 1D nature, the vorticity of this field profile is very homogeneous. Our work, in contrast, reports the regime when field is highly inhomogeneous and focuses on the chiral hotspots in the velocity field. This allows us to place chiral sources at such hotspots of maximal circular polarization, which enables a highly efficient one-way excitation at arbitrary frequency. In contrast to this, for the system considered in Ref. [44], the homogenized field becomes circularly polarized only in the limit of very large k , when $\tau \rightarrow k$, and $v_x \rightarrow \pm ikv_y$. Our approach thus offers a more efficient one-way excitation. To summarize this part, the two differences compared to [44] are the 2D nature of spin-locking in our system and the inhomogeneity of the velocity field vorticity with clear circularly polarized hotspots. To avoid any confusion, Fig. 1 (panel B) was revised and the text was changed accordingly.

Speaking of Ref. [45], their spin-momentum locking mechanism relies on a different type of field property, specifically resulting from the destructive interference of the phase-delayed reflective metasurface. Here, x-mirror symmetry is broken which generates a chiral velocity field wrapped around the structure. We stress that there is neither bulk topology nor any symmetry breaking property of our spin-momentum locking mechanism, and as such makes it markedly different from the method shown here.

Reviewer #2, specific remark 2

In the second scenario, the proposed chirality-based bulk mode propagation (Fig. 3) and directional edge emission (Fig. 4) have been well studied in previous works about acoustic valley-Hall phases and can be clearly explained by the valley-projected physics, such as valley pseudospin and valley-Hall topological edge states in Refs. [PRB 95, 174106 (2017); Nat. Phys. 13, 369 (2017); PRL 120, 246601 (2018)].

Authors’ response to the specific remark 2

We thank Reviewer #2 for their comment and for pointing out to these earlier relevant works, which we references in the revised manuscript. Nonetheless, for “the second scenario” we again

disagree with the reviewer's comments that "the proposed chirality-based bulk mode propagation and directional edge emission have been well studied in previous works", since the systems are clearly different. For instance, the noted reference [Phys. Rev. B 95, 174106 (2017)] report locking of orbital angular momentum to valley in a sonic crystal, but focuses on measuring this vorticity and beam splitting, and not directional excitation as in our work. Moreover, in our (clearly different) Kagome lattice, the circularly polarized nature of the modes manifests not only in the vicinity of the K and K' valleys, but along a wide range of momenta (see Fig. 2B). This is in addition to other differences, such as more degrees of freedom per unit cell in kagome lattice, which yields a richer band structure with three bands and chirality defined by the phase difference at three resonators constituting the unit cell. Similarly, Ref. [Phys. Rev. Lett. 120, 246601 (2018)] focuses on even more different aspects such as non-Hermiticity and valley physics in dimerized acoustic graphene (or acoustic hexagonal boron nitride), which is very different from kagome lattice studies here.

Finally, in Ref. [Nat. Phys. 13, 369 (2017)] selectively excited the valley edge states by changing the incident angle of near field source such that the projected momentum of the excited wave is tuned. However, in our work, we excite our topological edge states by circular-polarized field and we explicitly demonstrate the angular-momentum to linear momentum locking phenomenon. To our best knowledge, our work is the first one to experimentally reveal such exotic property of topological edge states in acoustics.

Although these works are different from ours, we find them relevant in broader context and, also because they are among ones of the seminal works in the topological acoustics, we cite them in the introduction of the revised manuscript.

Reviewer #2, minor/technical remarks and suggestions

Remark 1

1. In Fig. 1D, only one-dimensional spatial field distributions is provided. Sound propagations along the other direction in the x-y plane should also be discussed here to confirm the unidirectional excitation.

Authors' response

This remark is in line with that of Reviewer #1 and we believe that it was indeed important to emphasize the 2D nature of our system. To this aim, we verified the above 2D spin-locking in our system and plotted revised Fig. 1 and we show the 2-dimensional energy bands of the structure and the respective chirality of the mode. The revised Fig 1B shows chirality of the acoustic modes the 2D Brillouin zone via pseudo-spin vector field $\mathbf{S}=(S_x, S_y)$ where S_x is chirality in the Y-Z plane and S_y is chirality in the X-Z plane, defined as projections of the velocity field on circularly polarized basis in the respective planes.

Remark 2

2. Band diagram for the real acoustic lattice should be added into Fig. 2B to compare with the TBM results.

Authors' response

There is some confusion about this figure, as the band diagram was indeed obtained from first-principles simulation of realistic acoustic structure. We made this point clearer in the revised version of the manuscript, both in the text and in the figure caption.

Remark 3

3. During the measurements of the field distributions by using the chiral source, the authors describe the full amplitude response as a superposition of the contribution from each of the individual source sites. Why not use the several sources simultaneously? Is it a technical problem or is there any difference between these two methods please?

Authors' response

We thank Reviewer 2 for this comment. Due to the linear nature of our system, there is no difference between the two methods and they obviously yield identical result. However, this is perhaps an important point that we chose to clarify and we thank the reviewer for raising it. Why do we choose the superposition method over simultaneous excitation? There are two reasons: first, it is hard to fit multiple acoustic transducers into the structures without obstructing one another, especially for the 2-D array of holes when the chiral “hotspot” has a spatial volume on the order of mm^3 , which makes experiment more involved while not providing any additional information. We therefore chose to use tuning phase and superposing two measurements as its more effective in proof-of-principles demonstration of spin-locking. We have provided a clarification in the revised version of the manuscript in the methods section: “This method of superimposing several configurations of excitation was chosen over simultaneous placement of several sources to avoid undesirable obstruction of sound and formation of non-chiral standing wave patterns.”

Remaining minor/technical remarks and suggestions

4. Please have a double check on the writing, which includes some typos and misleading:

- 4.1) What do CCW and CW stand for?
- 4.2) Line 81: "...field. and this..." -> "...field. And this...".
- 4.3) A red dot is missed in Fig. 1B.
- 4.4) What is Jones vector?
- 4.5) Line 179: "G-K" -> "\Gamma-K".

Authors response

We thank the Reviewer for raising up these points and we have addressed them in the revised manuscript.

Reviewer #3:

General remarks by Reviewer #3

In the manuscript by Weiner et al., titled “Synthetic Pseudo-Spin-Hall Effect in Acoustic Metamaterials”, the authors investigate two types of synthetic pseudospins, which emerge either due to the evanescent nature of modes or due to symmetry of the lattice. While the first type of pseudospin was reported before, the authors bring novelty by considering inhomogeneous character of the nearfield in patterned metasurfaces. This feature is absent in modes localized due to excitation of internal degrees of freedom of materials (e.g., plasmons in metal) and in deep subwavelength limit considered so far. Therefore, I consider this result as a significant step forward, especially because the authors directly demonstrate that this property can be used for highly efficient directional excitation by placing circularly polarized sources at the chiral hotspots of the near-field. The second section, which focuses on the symmetry-engineered pseudo-spin, is also absolutely new to the best of my knowledge. While Kagome structures have been used to demonstrate topological properties and higher-order topological (corner) states by the authors, here they focus on a different aspect which largely evaded attention of the acoustic community. Thus, this work shows that both bulk and edge states can be directionally excited in both 2D (across the surface of the structure) and in 1D (along the edges) with very high efficiency.

The paper also very well-written and makes strong case for the use of synthetic pseudospins in acoustic metamaterials, which is likely to attract a significant interest from the Nature Communications readership. I there recommend this work for publication, after a minor correction.

Authors response to general remarks

We thank the Reviewer for the positive comments regarding our work and for recommending the manuscript for publication.

Reviewer #3, minor/technical remarks and suggestions

Remark 1

1) One small correction needed. In Fig 1C, LP should be replaced with LCP.

Authors' response

We thank Reviewer 3 for noticing this. The issue was corrected.

Remark 2

2) I also would like to ask for a clarification. From my point of view the system does not have to be topological to exhibit spin-Hall effect in the bulk. It would be good to clarify this point in the revision.

Authors' response

We fully agree with the Reviewer's comment. Regarding the two-dimensional array of holes, we stress that this chirality is an intrinsic property of the evanescent near-fields. Additionally, for the Kagome lattice, it is the lattice symmetry which guarantees the chirality of the modes, which is why chirality-guided propagation can exist within the bulk, as shown in Figure 3. We have clarified these points in the revised manuscript.

REVIEWERS' COMMENTS

Reviewer #1 (Remarks to the Author):

=====

Review of revised Nature Communications manuscript NCOMMS-22-12628A

"Synthetic Pseudo-Spin-Hall Effect in Acoustic Metamaterials".

The initial work was excellent, but it had many problems of presentation. I did list a very large number of problems (especially with the figures) that the authors had to fix. They did fix these, and their work improved the paper significantly.

Minor details that could be easily fixed:

- 1) The text writes units (like mm) sometimes in roman and sometimes in italics, at random.
- 2) Same with the use of subindices: should be roman for words and italics only for symbols.
- 3) The DOI of the references got messed up, for many papers, with random line breaks all over. Not clear why. Might be a formatting issue.

But these are small details that the authors (or the editors) could easily fix there.

The important issue is that this work is original, creative, interesting, systematic, well done, and now (after many improvements) ready for publication in Nat. Comm. (especially after fixing the minor details listed above).

Reviewer #2 (Remarks to the Author):

I want to thank the authors to address my comments. However, I am not convinced by their reply on the novelty of their work, especially about the second scenario. Although the authors did not mention the concept of valley physics in the entire manuscript, the directional excitation and the circular-polarized field-excited edge states, which are the main conclusions of the second scenario, have been well studied in valley-Hall topological photonics and acoustics. In the following, I will give my comments according to the authors' response about the differences of this work against the previous ones.

1. The authors said that, compared with Ref. [Phys. Rev. Lett. 120, 246601 (2018)], the structure used in this work, i.e. Kagome lattice, was different.

To my knowledge, the Kagome lattice-based valley-Hall phases have been studied in optics, elastics and acoustics, such as [Phys. Rev. B 100, 094303 (2019); ACS Photonics 7, 2089–2097 (2020); Phys. Rev. Research 2, 012011(R) (2020); arXiv: 1802.04404v1 (2018); EPL, 129, 44001 (2020)]. No matter which kind of model is used to study the valley-Hall phases, the valley-related physics or properties will not be affected.

2. The authors said that, compared with Ref. [Phys. Rev. B 95, 174106 (2017)], this work presented the directional excitation but not the beam splitting.

The directional excitation can be ascribed to the chiral valley bulk states, which can be obtained by the circular-polarized source, as well studied in both optics and acoustics, like Fig 2 in [Phys. Rev. B 96, 020202(R) (2017)], Fig 2 in [New J. Phys. 21, 093020 (2019)], Fig 2 in [Sci. Rep. 8, 1588 (2018)] and Fig 2 in [Phys. Rev. Lett. 120, 246601 (2018)].

3. The authors said that, compared with Ref. [Nat. Phys. 13, 369 (2017)], the topological edge states were excited by circular-polarized field.

Actually, due to the chiral properties of the valley states, it is known that not only the bulk states can be excited by the circular-polarized source as discussed in the point 2, but also the edge states can be triggered to propagate towards the specific direction [Sci. Rep. 8, 1588 (2018); New J. Phys. 21, 093020 (2019); EPL, 129, 44001 (2020)].

As a result, I still do not find enough novel physics and application potential in acoustics after carefully reading the response letter. Unfortunately, I cannot recommend it for publication in this high-impact journal.

Reviewer #3 (Remarks to the Author):

The authors have addressed the referees' remarks, and I recommend publication of this manuscript in its present form.

AUTHORS' RESPONSE TO REVIEWERS' COMMENTS

Reviewer #1

General remarks

The initial work was excellent, but it had many problems of presentation. I did list a very large number of problems (especially with the figures) that the authors had to fix. They did fix these, and their work improved the paper significantly.

Authors' response to general remarks

We appreciate positive response from Reviewer #1 to our work and for their valuable suggestions which allowed us to improve the manuscript.

Minor points for revision

- 1) The text writes units (like mm) sometimes in roman and sometimes in italics, at random.
- 2) Same with the use of subindices: should be roman for words and italics only for symbols.
- 3) The DOI of the references got messed up, for many papers, with random line breaks all over. Not clear why. Might be a formatting issue.

But these are small details that the authors (or the editors) could easily fix there.

Authors' response to minor points from Reviewers #1

We thank Reviewer #1 for their very careful attention to details and we fixed the listed issues. Units and subscripts were unified in formatting in accordance with the Nature Communications guidelines. The list of references was recreated from scratch too.

Reviewer #1 concluding remarks

The important issue is that this work is original, creative, interesting, systematic, well done, and now (after many improvements) ready for publication in Nat. Comm. (especially after fixing the minor details listed above).

Authors' response to concluding remarks

We thank the Reviewer #1 again for their careful reading and for all the helpful suggestions which allowed us to significantly improve the quality of the work. We also appreciate their recommendation to publish our work in Nature Communications.

Reviewer #2

Reviewer #2 general remarks

I want to thank the authors to address my comments. However, I am not convinced by their reply on the novelty of their work, especially about the second scenario. Although the authors did not mention the concept of valley physics in the entire manuscript, the directional excitation and the circular-polarized field-excited edge states, which are the main conclusions of the second scenario, have been well studied in valley-Hall topological photonics and acoustics. In the following, I will give my comments according to the authors' response about the differences of this work against the previous ones.

1. The authors said that, compared with Ref. [Phys. Rev. Lett. 120, 246601 (2018)], the structure used in this work, i.e. Kagome lattice, was different.

To my knowledge, the Kagome lattice-based valley-Hall phases have been studied in optics, elastics and acoustics, such as [Phys. Rev. B 100, 094303 (2019); ACS Photonics 7, 2089–2097 (2020); Phys. Rev. Research 2, 012011(R) (2020); arXiv: 1802.04404v1 (2018); EPL, 129, 44001 (2020)]. No matter which kind of model is used to study the valley-Hall phases, the valley-related physics or properties will not be affected.

2. The authors said that, compared with Ref. [Phys. Rev. B 95, 174106 (2017)], this work presented the directional excitation but not the beam splitting.

The directional excitation can be ascribed to the chiral valley bulk states, which can be obtained by the circular-polarized source, as well studied in both optics and acoustics, like Fig 2 in [Phys. Rev. B 96, 020202(R) (2017)], Fig 2 in [New J. Phys. 21, 093020 (2019)], Fig 2 in [Sci. Rep. 8, 1588 (2018)] and Fig 2 in [Phys. Rev. Lett. 120, 246601 (2018)].

3. The authors said that, compared with Ref. [Nat. Phys. 13, 369 (2017)], the topological edge states were excited by circular-polarized field.

Actually, due to the chiral properties of the valley states, it is known that not only the bulk states can be excited by the circular-polarized source as discussed in the point 2, but also the edge states can be triggered to propagate towards the specific direction [Sci. Rep. 8, 1588 (2018); New J. Phys. 21, 093020 (2019); EPL, 129, 44001 (2020)].

Authors' response to Reviewer #2 general remark

We thank Reviewer #2 again for sharing their opinion regarding analogies between our work and valley Hall-effect topological structures which were previously studied, include the ones they listed. However, we believe there is again misunderstanding of the whole concept of our work and its interpretation is not what we intended too. On the other hand, we are happy to see that the Reviewer #2 no longer questions originality of our results in the first scenario. However, the whole point of our work is to demonstrate the universality of the spin-Hall effect and in one work show the possibility to use two distinct mechanisms to generate and control sound by pseudo-spins.

Moreover, we intentionally refrain from using the language of valley-Hall effect interpretation, typically given from the point of view of valley-Hall Chern number, and, in our work we focus more on the nature of the pseudo-spin and bulk properties rather than edge properties studies before in the context of the acoustic valley-Hall topological boundary states.

Our desire to refrain from valley-Hall language is related to the fact that the kagome system is not a good valley Hall insulator and this is evident from the gapped character of the edge states in Fig. 4. The treatment proposed by us is based solely on symmetry and a more appropriate topological invariant - Wannier type bulk polarization [Nature materials 18 (2), 113-120 (2018)] rather than valley Chern number used in valley-Hall topological systems.

We would like to point out that we are well-familiar with prior works on kagome structures, and, in fact, our group was the first to investigate kagome lattices in acoustics [New Journal of Physics 19 (5), 055002 (2017)] in the context of spin-polarized edge states and, later, higher-order corner states [Nature materials 18 (2), 113-120 (2018)], as well as in photonics.

We therefore kindly disagree with the Reviewer #3 in their opinion about novelty of our work. In fact, our work is the first one to rigorously study the nature of one-way edge states in acoustic kagome lattices, with the symmetry-based approach with the lattice symmetry operators considered as generators of a pseudo-spin both in the bulk and on the edges, which is in contrast to rather superficial language of valley-Hall effect given in the references listed by the Reviewer #2.

We therefore insist that our system is not a valley-Hall topological insulator but a higher-order topological insulator with Wannier-type polarization in the bulk, whose symmetries manifest in the spin-momentum locking of bulk modes and edge states, and which only crudely resembles the valley-Hall topological phase.

We hope the Reviewer #2 will understand the difference between our work and previous studies of acoustic kagome lattices. We point out once again that no rigorous study of bulk spin-Hall effect was reported prior to our work and only shallow interpretation in terms of valley-Hall effect was given for the edge states.

In revised paper we added the following paragraph to explain this point:

“We note that kagome lattices are sometimes considered as valley-Hall insulator, which is rather superficial treatment, because valley-Chern number is not a good topological invariant and Wannier bulk polarization should be used instead. Although the effects of spin-momentum locking reported here can be found in other topological systems, including true valley-Hall systems such as acoustic graphene, the difference is clearly seen in the strongly gapped nature of the edge states, which indicates that the respective bulk boundary correspondence, that would manifest in valley-Hall systems in the form of gapless edge states, does not hold for kagome lattices.

Topological edge states due to the nonvanishing bulk polarization in topological domain emerge at the boundary between topologically distinct crystals, and they have remarkable properties associated with their topological nature.”

We hope that Reviewer #2 will agree with our point and will change his/her opinion about our work.

We also note that, if the editor permits to exceed the limit on number of references, we will be more than happy to cite references listed by the Reviewer #2 in the proper context.

Reviewer #2 concluding remark

As a result, I still do not find enough novel physics and application potential in acoustics after carefully reading the response letter. Unfortunately, I cannot recommend it for publication in this high-impact journal.

Authors' response to Reviewer #2 concluding remark

We appreciate all valuable recommendations from Reviewer #2 which, we believe were crucial for improving the quality of our work.

We also note that following Reviewer #2 advice, in the latest revision of our work we used our new custom setup based on two sources oscillating with controllable phase shift, which allowed us to generate circularly polarized acoustic fields near hotspots in scenario 1 and to demonstrate directional emission of sound in one measurement (instead of taking superposition of two measurements as was done before). One-way excitation was also demonstrated for two directions along the structure (unlike the case of one direction in the original and second revisions of the paper). The Fig. 1 was revised accordingly and the results for the additional G-M direction were added to the supplement.

We hope that, in the end, the Reviewer #2 will agree with the findings of the other two Reviewers.

Finally, we thank Reviewer #2 for their patience and for their continuous and careful attention to details of our work.

Reviewer #3

Reviewer #3 remark

The authors have addressed the referees' remarks, and I recommend publication of this manuscript in its present form.

Authors' response to Reviewer #3 remark

We thank Reviewer #3 for noticing our substantial revision to the manuscript aimed at addressing the remark by all the Reviewers. We very much appreciate their recommendation to publish our work in the present form.